# T3V2V: Test Time Training for Domain Adaptation in Video-to-Video Editing

## Abstract

*In the realm of generative AI, state-of-the-art Video-to-Video (V2V) editing models can perform diverse edits based on different conditions and generate new videos. Despite their ability to generate various video edits, these models still face significant frame inconsistencies, such as motion discrepancies and unnatural background changes. This paper addresses these issues by analyzing video inconsistencies through domain shifts and implementing domain control based on this theory. Furthermore, a test-time compute-optimal sampling method for better representation of different video domains is proposed, which is a high-performance test-time training (TTT) method. By leveraging this TTT method, we propose **T3V2V** (TTT-V2V editing). Our method utilizes frame-level information to establish an unsupervised TTT learning process, providing more precise guidance for the image-to-video (I2V) generation process and enhancing video consistency through effective self-supervised parameter optimization and domain adaptation. Extensive experiments on the DAVIS-EDIT benchmark show that T3V2V outperforms previous state-of-the-art models. The self-supervised nature of our TTT approach further enables robust generalization to diverse V2V editing tasks, establishing a new paradigm for V2V synthesis.*

## 1. Introduction

Video-to-Video (V2V) editing has gained increasing attention due to its potential to create various meaningful video content based on existing videos. Video-to-Video (V2V) editing techniques have made significant progress, evolving from traditional optical flow-based methods [7, 13] to GAN-based [5, 23] and Transformer-based [29] methods, with diffusion-based approaches now leading the field [6, 14–16]. State-of-the-art diffusion-based V2V editing models [14, 16] enabled multiple editing approaches, such as text prompts, reference images, style guidance, allowing for more flexible video modifications. These modifications rely on the first-frame editing technique [23] and the Image-to-Video (I2V) generation model [32], where modifications

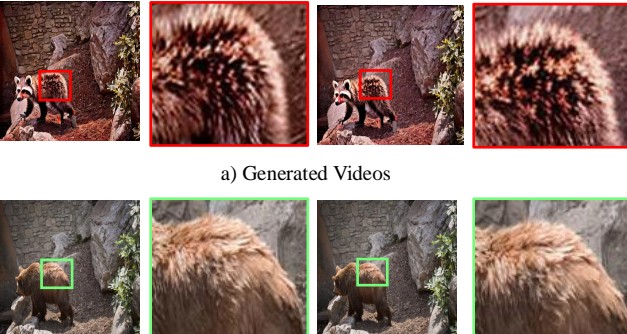

a) Generated Videos

b) Target Videos

Figure 1. State-of-the-art (SOTA) video-to-video edited video (up) and original video (down).

are applied to the initial frame and subsequently propagated throughout the entire video.

However, while these first-frame editing approaches allow for explicit and efficient video editing, they may inevitably suffer from frame inconsistencies, including object motion discrepancies and unnatural background. Since the editing of subsequent frames relies solely on the initial frame edit without explicitly modeling the video dynamics, the object motion and background-changing in the edited video may not align with the natural state, thus leading to visual instability in the edited videos. Despite the fact that the state-of-the-art frameworks achieved realistic video edits, obvious inconsistencies can still be observed in the edited videos. As illustrated in Figure 1, we magnified the same position in two consecutive frames for a detailed comparison. It is evident that the generated video exhibits noticeable temporal discontinuities between adjacent frames, whereas the target video demonstrates a high degree of temporal coherence. To address these limitations, we analyze frame inconsistencies through domain shifts and present a self-supervised improvement approach from the perspective of domain control and adaptation, which learns from the frame-level information and provides more precise guidance to the I2V generation process.

Therefore in this paper, we introduce a test-time compute-

optimal sampling strategy, which is an efficient Test-Time Training (TTT) [21] method tailored for V2V generation models, allowing the model to adapt to each video individually and thus enhance frame consistencies. TTT [21] is a dynamic parameter adjustment mechanism during inference time to adapt the model to unforeseen data distributions with domain shifts, which can also be regarded as a one-shot domain adaptation approach [10]. It usually employs an extra self-supervised learning task to assist the model's main task in fitting with new data distributions and converging to a lower loss. We make the following key contributions in this work:

- We established a new view for V2V editing from the theory of domain shift.
- We introduce a novel TTT method for diffusion-based video generation, a test-time compute-optimal sampling strategy. Based on this, we propose TTT-V2V editing model, termed **T3V2V**.
- We conducted extensive experiments on the DAVIS-EDIT benchmark [16] and compared the performance of T3V2V with state-of-the-art V2V frameworks. T3V2V outperforms the state-of-the-art frameworks, demonstrating the effectiveness of our method.

## 2. Related Work

### 2.1. Video-to-Video Editing Models

Video-to-Video (V2V) editing models are designed to modify video content and generate new videos, enabling tasks such as style transfer, object replacement, text-prompted editing, etc. Early methods often relied on optical flow-based warping and GAN-based generation, which might be only effective for specific tasks and struggle with generalization. With the rise of the diffusion model, video editing has been significantly enhanced by its powerful generative ability. The main existing video generation and V2V editing methods have the following types: 1) Frame-wise editing methods [1, 4, 24] apply image-based diffusion models to each video frame independently, followed by post-processing techniques such as optical flow warping to improve temporal consistency; 2) End-to-End diffusion methods [9, 12, 20] train a unified video diffusion model that directly generates or edits entire video sequences; 3) Fine-tuning-based methods [18, 27, 31] adapt pre-trained diffusion models to video editing by fine-tuning on a specific video and other information while preserving temporal consistency ; 4) First-frame-based methods [14–16, 19] edit the first frame and propagate modifications across the sequence by transferring motion from the source video. Although most V2V models still focus on specific tasks, state-of-the-art V2V models [14, 16] offer versatile and comprehensive editing frameworks, effectively supporting a wide range of high-quality editing tasks. These models leverage

existing image editing methods (e.g., Anydoor [8], Instruct-Pix2Pix [3]) to edit the first frame. Then they integrate the Image-to-Video (I2V) generation model [32] to propagate edits across frames and transfer the original motion to the new videos. However, the video edits still exhibit inconsistencies between frames and unnaturalness, with limitations in preserving the fine details.

### 2.2. Test-Time Training

Test-Time Training (TTT) is a technique that adjusts the model to fit different data domains by integrating the extra self-supervised learning process during inference time [21], which can also be considered as a one-shot domain adaptation approach. TTT adapts the model to fit different data distributions and mitigates the influence of domain shifts by learning data distribution representations from the test sample. Recently, TTT has gained increasing attention and has been successfully applied to a variety of machine learning tasks [2, 10, 11, 22, 26]. For example, Gandelsman et al. selected masked autoencoder (MAE) reconstruction as the self-supervised TTT task and observed significant improvements in the recognition task [10]. Leveraging self-supervised adaptation, CustomTTT [2] introduces TTT framework for motion and appearance-customized text-to-video generation. The choice of the test-time training approach and the main task are of vital importance, as they will impact the test sample's representation of the test data distribution domain and overall optimization quality. In this paper, we extend the test-time training (TTT) setting to the Video-to-Video (V2V) generative task and propose the test-time compute-optimal sampling method, aiming to enhance the quality of the edited video by controlling the domain shift.

## 3. Preliminary

### 3.1. Domain shifts control

Test-Time Training (TTT) allows for updating pre-trained models during deployment to handle changing data distributions. Due to uncontrollable domain shifts, TTT is prone to error accumulation, leading to blurry object boundaries between the target and original videos during the video generation process. To address this issue, we propose an improved method to suppress domain shifts. Specifically, we computes domain shift for each image domain reconstructed at every step $t$ of the diffusion model generation process. Here, we use $\mathbf{D}_t^{Shift}$ to describe the domain shift at timestamp $t$, construct $\mathbf{D}_t^{tar}$ to represent the target domain at timestamp $t$ (which denotes the edited target image features), and employ $\mathbf{D}_t^{sour}$ to indicate the source domain at timestamp $t$ (which represents the original image features).

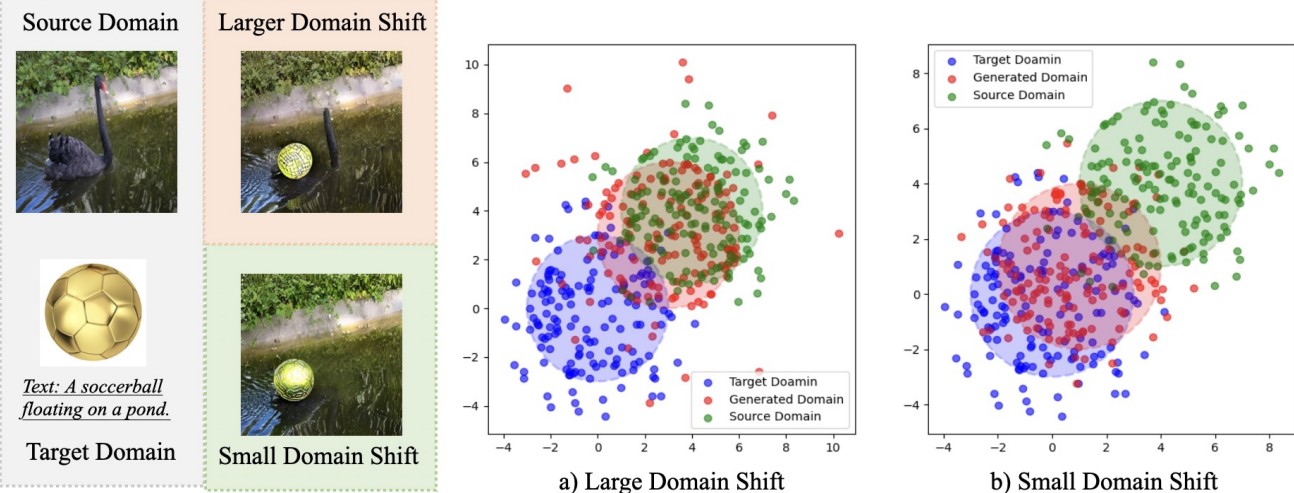

Figure 2. Visualization of domain shifts control. Here, different 2048-dimensional image feature tensors from different images with large domain shift and small domain shift are extracted using Inception-V3. To realize better visualization, image feature tensors are mapped to a 2D space using the t-SNE approach and shown in a) and b).

$$\mathbf{D}_t^{Shift} = MSE(\mathbf{D}_t^{tar} - \mathbf{D}_t^{sour}) \quad (1)$$

We visualize the image embeddings with and without domain shift control. As illustrated in Figure 2 (a), in the absence of domain shift enhancement, the generated image features exhibit a strong bias towards the source domain, leading to a dispersed object distribution. Conversely, in Figure 2 (b), where domain shift control is applied, the image features exhibit a closer alignment with the target domain, demonstrating a more coherent object distribution that is consistent with the intended image editing target. This highlights the effectiveness of our proposed domain shift control in ensuring that the generated content more accurately reflects the target domain.

### 3.2. Test-time compute-optimal sampling

To further prevent our model from over-optimizing during the test-time training process, while also optimizing the test-time computational budget in the video editing process, we propose a reward-aligned target noise distribution sampling method. This method is specifically designed for diffusion sampling, and while it maintains the ability to follow textual instructions, it also enables consistency control in video editing. Based on this sampling feature, we performed Domain distribution fitting and defined the source domain as:

$$\mathbf{D}_t^{sour} = \min_D \mathbb{E}_{x \sim D}[L(x : \mathbf{I}_t^{tar})|\mathbf{I}_{0 \to t-1}^{tar}, \mathbf{I}_0^{sour}, t] \quad (2)$$

Here, we use target images $\mathbf{I}_{0 \to t-1}^{tar}$ generated from previous timesteps $0 \to t-1$ as one of the prior conditions,

while incorporating source images $\mathbf{I}_0^{sour}$ from the initial timestep and the current timestep $t$ as additional priors to perform source distribution generation at timestep $t$.

Similarly, we define the target domain as a distribution that closely aligns with the desired output characteristics, which can ensure that the model effectively adapts to the target distribution while maintaining computational efficiency during the video editing process. By leveraging the reward-aligned target noise distribution sampling, we aim to optimize the ability of our model to generate content that adheres to the specific editing intent. This alignment not only enhances the consistency of the generated content but also facilitates seamless integration of domain shift control, ensuring that the performance of the model is tailored to the target domain without overfitting.

$$\mathbf{D}_t^{tar} = \min_D \mathbb{E}_{x \sim D}[L(x : \mathbf{I}_t^{tar})|\mathbf{I}_{0 \to t-1}^{tar}, guidance_t, t] \quad (3)$$

In the representation of target domain, we use target images $\mathbf{I}_{0 \to t-1}^{tar}$ generated from previous timestamps $0 \to t-1$ as one of the prior conditions, while incorporating target guidance embedding $guidance_t$ from the timestamp $t$ and the current timestamp $t$ as additional priors to perform target distribution generation at timestamp $t$. In our problem setting for video-to-video, the target motion embedding is given by the pre-trained large language model, and thus the edited video with guidance term enforcing the diffusion process to stay close to the pre-trained diffusion process.

## 4. Methodology

After completing the previous introduction to the underlying theory, we will give a brief overview of the overall video-to-video framework and introduce our key components and specific test-time training settings. The overall video-to-video framework is shown in Figure 3, where we first edit the first frame and generate an image sequence based on it. In this process, we perform optical flow estimation and extract objects for each frame, and optimize frame consistency based on the estimated optical flow and the extracted objects. At the same time, we introduced the concept of spatial understanding to enhance the spatial understanding of the model for each object and scene by performing depth-based estimation of all scenes and objects in the visible domain and using the previously mentioned objects extracted for each frame as the object shape prior condition for spatial understanding. Finally, we use test-time training to optimize the generalization performance of TTT video-to-video editing (T3V2V) over different image domains.

### 4.1. Frame Consistency Maintaining

In T3V2V, we first use the InstructPix2Pix [3] method to edit the first frame of the image based on the given external prompt, and then we combine it with the proposed enhancement method to generate the subsequent frames. In the process of generating subsequent frames, we first need to ensure the consistency of the frame. Considering that the most important point in frame consistency alignment is the consistency and perception of the frame body, here we first use a pretrained multi-modal large language model to judge the main component of the first frame before and after editing, and then extract it. Specifically, we first use the Florence-2 [28] model for frame understanding, and then we use Grounding-dino [17] for open-vocabulary object segmentation to extract the segmentation masks $\{\mathbf{M}_0^{sour}, \mathbf{M}_1^{sour}, ..., \mathbf{M}_k^{sour}\}$ of the main component in different source frames. It is worth noting that here, in order to migrate such motion information to the edited target frame, we perform the same open-vocabulary object segmentation based on the edited first frame to get $\mathbf{M}_0^{tar}$.

After that, we use the optical flow based on SEA-RAFT [25] to represent the consistency information of the main component, as well as to control the coherence of the motion between frames. More detailed, with the input source frames $\{\mathbf{I}_0^{sour}, \mathbf{I}_1^{sour}, ..., \mathbf{I}_T^{sour}\}$, we can get optical flows $\{\mathbf{O}_0^{sour}, \mathbf{O}_1^{sour}, ..., \mathbf{O}_T^{sour} - 1\}$ between frames, in which $\mathbf{O}_t^{sour}$ = SEA-RAFT $(\mathbf{I}_t^{sour}, \mathbf{I}_{t+1}^{sour})$. Combined with the segmentation masks obtained earlier, we extract the optical flow of the main component in the source video, and superimpose the optical flow values at the position in each frame to construct an optical flow-based motion function to evaluate its average motion state, and realize the evaluation of the degree of motion intensity.

$$\mathbf{O}_t^{\text{motion}}(x,y) = \begin{cases} \frac{1}{\mathbf{M}_t} \sum_{(x,y) \in \mathbf{M}_t} \mathbf{O}_t^{sour}(x,y), \\ \qquad \text{where} \quad (x,y) \in f_{paste}(\mathbf{M}_t^{tar}) \\ 0, \qquad \text{otherwise} \end{cases} \tag{4}$$

where $(x,y)$ denote the pixel located at the $x$-th row and the $y$-th column of $\mathbf{M}_t$. Then, we calculate the consistency metric within the main component area of edited frame through optical flow-based motion function $\mathbf{O}_t^{\text{motion}}(x,y)$ on $\mathbf{M}_k^{tar}$. Based on such an optical flow-based motion function, we can obtain a coherent representation in the edited frame, which is formulated as follows:

$$\mathbf{CON}_t(x,y) = \mathbf{O}_t^{sour} \otimes (1 - \mathbf{M}_t^{tar}) + \mathbf{O}_t^{\text{motion}}(x,y) \tag{5}$$

Here, $\otimes$ means that the tensor is multiplied one by one according to the index. After the computation of the previous frame, we can estimate the mask of the main component of the next frame based on a mixture of open-vocabulary and warping methods, and then we can realize the mask of the main component of each frame based on the mask of the edited target frame, and then realize the consistency maintaining of the main component in each frame.

### 4.2. Spatial Understanding

After completing the optimization for maintaining subject consistency in different frames, we also need to consider the consistency of the scene, as well as the consistency of the relative positions of the scene and objects throughout the video. This requires our model to have a good spatial understanding of the region where the video was taken. In recent years, with the development of pre-trained monocular depth estimation models [30], existing large models are able to achieve better monocular depth estimation. We utilize this capability of existing models to provide a strong spatial prior $\mathbf{D}^{sour} = \{\mathbf{D}_0^{sour}, \mathbf{D}_1^{sour}, ..., \mathbf{D}_T^{sour}\}$ for our video model. Similar to the previous estimation of the average optical flow-based motion function, here we also compute the average depth of the main component throughout the process, which is used for the estimation of the position in the frame.

$$\mathbf{D}_t^{\text{pos}}(x,y) = \begin{cases} \frac{1}{\mathbf{M}_t} \sum_{(x,y) \in \mathbf{M}_t} \mathbf{D}_t^{sour}(x,y), \\ \qquad \text{where} \quad (x,y) \in \mathbf{M}_t^{tar} \\ 0, \qquad \text{otherwise} \end{cases} \tag{6}$$

where $(x,y)$ represents the pixel at the $x$-th row and $y$-th column. Finally, we construct the $k$-th simulated depth map $\mathbf{D}_t^{tar}$ using following function:

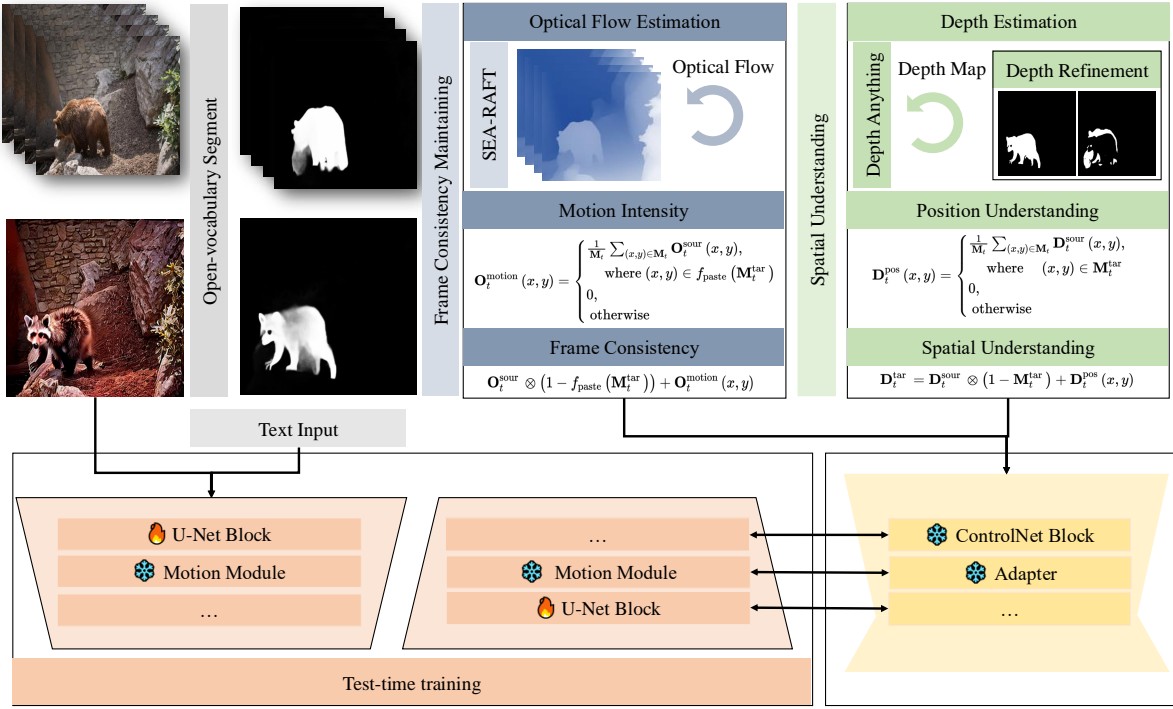

Figure 3. Overall framework of T3V2V.

$$\mathbf{D}_t^{tar} = \mathbf{D}_t^{sour} \otimes (1 - \mathbf{M}_t^{tar}) + \mathbf{D}_t^{pos}(x, y) \quad (7)$$

Similarly, by conducting a step-by-step reasoning of this depth inference approach, we can obtain depth estimates for each frame of the edited content, thereby enhancing the spatial understanding of T3V2V. It is important to note that we have observed instances where the depth estimates for the edited frames may be incomplete. To address this issue, we adopt the depth completion method from StableV2V [16], which utilizes completion networks to repair optical flows, and we propose a depth refinement network based on this paradigm.

$$\mathbf{M}_{refine} = f_{paste}((1 - \mathbf{M}_t^{tar}) \otimes \mathbf{M}_t^{sour})$$
$$\mathbf{D}_{refine}^{tar} = \mathbf{M}_{refine} \otimes \mathbf{D}_{refine}^{tar} \quad (8)$$

In this paradigm, we reuse the shape mask $\mathbf{M}_t^{tar}$ in source video at timestamp $t$ to ensure the spatial understanding of depth refinement. More specifically, as shown in the above equation, we extract the part of the source video that removes the main component $1 - \mathbf{M}_t^{tar}$ and conduct an element-by-element multiplication $\otimes$ with the mask in the target video $\mathbf{M}_t^{sour}$, so that we can get the regions that may have missing depths $\mathbf{M}_{refine}$. After that, we use the calculated regions that may have missing depths to perform depth

redefinition and get the depth refinement result $\mathbf{D}_{refine}^{tar}$. We input the combined data from the estimated depth maps, the refined motion information, and the initial edited content into the shape-guided refinement network. This process yields the final depth maps, which provide accurately simulated depth information for the edited video. These refined depth maps are crucial for providing precise guidance in the context of content generation.

## 4.3. Test-time Training

In the previous preliminary, we mentioned a key issue in video-to-video editing, i.e., there is likely to be a very large domain gap between the edited video and the source video, and at the same time, the user's text inputs may be diverse, so how to realize unsupervised test-time training is very important for high-quality video editing. At the same time, considering the inference speed limitation of the video editing model, only updating the downstream image to video component can achieve the most economical domain gap optimization. More specifically, we freeze the weights associated with optimizing Frame Consistency Maintenance and Spatial Understanding to prevent the model from experiencing catastrophic forgetting during domain adaptation. At the same time, we unfreeze the weights of the U-Net Block and perform self-supervised updates based on the calculated Domain Shift. This approach enables the model to

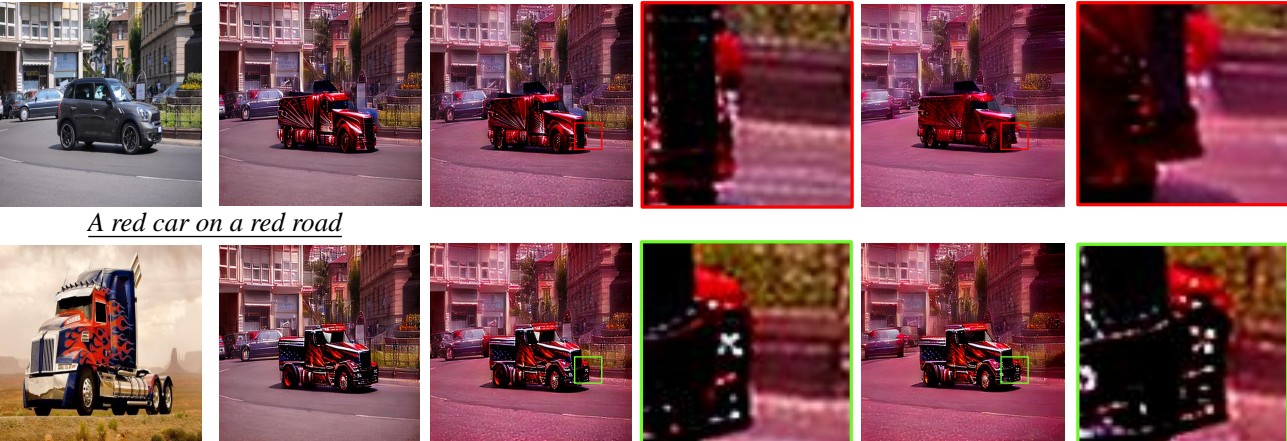

Figure 4. Generated video case using StableV2V (up) and T3V2V (down).

better comprehend various video-to-video editing tasks, as is shown in Figure 3. For an efficient domain shift calculation, we propose a sampling method shown in Algorithm 1.

---

**Algorithm 1** Efficient Sampling for Test-time Training

---

1: **Input:** Target frame $\mathbf{I}_t^{tar}$, $\mathbf{I}_{0 \to t-1}^{tar}$, $\mathbf{I}_0^{sour}$, $t$, $guidance_t$
2: **Output:** $\mathbf{D}_t^{tar}$, $\mathbf{D}_t^{sour}$.
3: **for** each denoising step $k$ in diffusion **do**
4:      Calculate $L_k^{MSE}(\mathbf{I}_k^{tar}, guidance_k)$.
5:      2×Upsample to get $x_t^{tar} \sim \mathbf{D}_t^{tar}$, where:
6:          $\arg\max_{x \in \mathbf{I}_t^{tar}} \frac{\delta L_k^{MSE}}{\delta x}$.
7:      Calculate $L_k^{MSE}(\mathbf{I}_k^{sour}, \mathbf{I}_0^{sour})$
8:      2×Upsample to get $x_t^{sour} \sim \mathbf{D}_t^{sour}$, where:
9:          $\arg\max_{x \in \mathbf{I}_t^{sour}} \frac{\delta L_k^{MSE}}{\delta x}$.
10: **end for**

---

In this sampling algorithm, we feed Target frame $\mathbf{I}_t^{tar}$, $\mathbf{I}_{0 \to t-1}^{tar}$, $\mathbf{I}_0^{sour}$, $t$, $guidance_t$ into it, and then get the sampled results $\mathbf{D}_t^{tar}$, $\mathbf{D}_t^{sour}$. Notably, we have introduced a gradient-based upsampling method that accelerates the iterative optimization of the loss while reducing the computational burden associated with gradient calculations. Additionally, as described in the preliminary section, our evaluation of domain shift is also conducted using the MSE function, ensuring that this sampling process does not introduce excessive computational overhead.

## 5. Experiment

### 5.1. Experimental Setup.

To validate the superiority of our proposed T3V2V method, we need to compare the generated videos against the state-of-the-art (SOTA) methods. For this purpose, we have se-

lected the previous generation SOTA model, StableV2V, as our benchmark. Additionally, we examine and evaluate text-based video editing techniques, generating edited videos by modifying only the input guiding texts. In order to maintain a fair assessment, we used the DAVIS-EDIT data mentioned in the StableV2V article as the test data, and more specifically, we took 100 sets of test cases from DAVIS-EDIT-C, which involves changing objects. Finally, we compare our method with StableV2V to benchmark its performance, and use subjective and objective indicators to realize a broadly comparison.

**Subjective indicators. Consistency** evaluates the uniformity of visual elements, pacing, and technical parameters throughout a video, identifying abrupt shifts in quality, lighting, or style that disrupt viewer immersion. **Fluency** complements this by assessing the smoothness of motion, transitions, and playback, ensuring no stuttering, frame drops, or temporal disruptions like choppy edits or audio-visual mismatches. Finally, **Naturalness** emphasizes the authenticity of the video's replication of real-world experiences, avoiding over-processing artifacts or unrealistic color grading that undermine believability. Together, We used prompt engineering to promote GPT-4 to achieve fair evaluation of different metrics, but also standard consistent alignment of human raters. As shown in Figure 4, we present a editing case of StableV2V and T3V2V for the same input conditions. We selected frames from the same moment in two videos and magnified the key detail area. It can be observed that, StableV2V has limited performance in terms of overall style transfer and detail preservation, while T3V2V achieves better results, making the object closer to the target. Additionally, in terms of video coherence, T3V2V produces a more smoother effect, whereas the inconsistency in the background and the lack of details make the results of StableV2V appear less natural.

Table 1. Comparison of subjective indicators

| Method | Evaluator | Naturalness (↑) | Consistency (↑) | Fluency (↑) |
|---|---|---|---|---|
| StableV2V | GPT-4 | (78/100) | (75/100) | (84/100) |
| | Human | (76/100) | (76/100) | (80/100) |
| **T3V2V (Ours)** | GPT-4 | **(89/100)** | **(90/100)** | **(87/100)** |
| | | (↑ 14%) | (↑ 20%) | (↑ 4%) |
| | Human | **(87/100)** | **(92/100)** | **(88/100)** |
| | | (↑ 14%) | (↑ 21%) | (↑ 10%) |

Table 2. Comparison of Objective indicators

| Method | FVD (↓) | CLIP Score (↑) | DOVER (↑) |
|---|---|---|---|
| StableV2V | (13.98/16.92) | (25.56/25.62) | (67.88/70.95) |
| **T3TV2V (Ours)** | **(12.56/15.98)** | **(26.34/26.96)** | **(67.92/71.14)** |

**Objective indicators.** Here we use the Frechet Video Distance (FVD), CLIP score and DOVER (DOmain-aware VidEo Relevance measure) metrics for objective assessment. **FVD** measures video realism by comparing synthesized and real video distributions using features from a pretrained 3D network. **CLIP score** assesses text-video alignment via vision-language embedding cosine similarity. **DOVER** integrates content fidelity and motion coherence through spatial-temporal fusion and domain-aware contrastive learning.

### 5.2. Comparison with advanced methods.

As mentioned earlier, we tested the performance of the videos generated by StableV2V and T3V2V under various evaluation metrics, with the final results shown in Tables 1 and Table 2. The results show that T3V2V outperforms StableV2V in terms of naturalness, consistency, and fluency. Notably, in the consistency metric, T3V2V shows an improvement of nearly 20% compared to the previous state-of-the-art (SOTA). This improvement highlights that T3V2V has made significant progress in maintaining the coherence between the generated video content and the input information. In addition, T3V2V also demonstrates superior performance in objective evaluations. These excellent evaluation results indicate that T3V2V not only enhances the subjective experience but also performs significantly better in quantitative metrics compared to StableV2V.

Besides, here we present an editing case of StableV2V and T3V2V for the same input conditions shown in Figure 4. We selected frames from the same moment in two videos and magnified the key detail area. It can be observed that, StableV2V has limited performance in terms of overall style transfer and detail preservation, while T3V2V achieves better results, making the object closer to the target. Additionally, in terms of video coherence, T3V2V produces a more smoother effect, whereas the inconsistency in the background and the lack of details make the results of StableV2V appear less natural.

## 6. Conclusion

In this work, we address the critical challenge of frame inconsistency in Video-to-Video (V2V) synthesis by introducing **T3V2V**, a novel test-time training framework grounded in domain shift analysis and adaptive domain control. Our method leverages self-supervised test-time training (TTT) to align video domains dynamically, mitigating motion discrepancies and unnatural artifacts through compute-optimal sampling and frame-level unsupervised learning.T3V2V achieves superior spatial-temporal coherence compared to existing state-of-the-art models, as validated by extensive experiments on the DAVIS-EDIT benchmark. The TTT design enables robust generalization, bridging conditional synthesis with real-world consistency. In the future, we will investigate real-time optimization strategies for dynamic, user-driven editing scenarios.

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
