# OpenReview forum: "T3V2V: Test Time Training for Domain Adaptation in Video-to-Video Editing"
_thecvf.com/CVPR/2025/Workshop/CVEU — CVPR 2025_

### Official Review · Reviewer_uSt9 · 2025-03-14
**T3V2V: Test Time Training for Domain Adaptation in Video-to-Video Editing**

**Rating:** 4
**Confidence:** 4

**Review:**

Contribution:
1.The paper introduces a framework for Video-to-Video editing that addresses frame inconsistency issues by leveraging domain shift analysis and adaptive domain control through test-time training (TTT).
2.It proposes a test-time compute-optimal sampling strategy to dynamically adjust model parameters during inference, enhancing video consistency and reducing motion discrepancies and unnatural artifacts.
3.Experiments on the DAVIS-EDIT benchmark demonstrate that T3V2V outperforms state-of-the-art model.

Cons:
1. It is suggested that the authors discuss the computational overhead of the TTT approach, as the authors said that it is an efficient method. TTT may increase the computational burden, especially when dealing with long videos.
2. The comparative experiments in the paper focus on comparisons with StableV2V,  but lacks comparisons with other types of V2V editing methods. This may result in an incomplete assessment of the performance of T3V2V.
3. Experimental Setup: The paper uses the GPT-4 and human assessments as subjective assessment metrics, but does not detail the specific processes and criteria for the assessments. For example, how do you ensure that GPT-4 assessments are consistent with human assessments?

---

### Official Review · Reviewer_6GDp · 2025-03-22
**Review of Test Time Training for Domain Adaptation in Video-to-Video Editing**

**Rating:** 5
**Confidence:** 4

**Review:**

This paper introduces T3V2V, a framework to address challenges in Video-to-Video (V2V) editing especially in scenarios such as unnatural background changes. The issues highlighted are common in state-of-the-art V2V editing models and proposed approach involves analyzing video inconsistencies through the concept of domain shift and implementing domain control. Authors proposed a test-time training (TTT) method with a compute-optimal sampling strategy to better represent different video domains. The proposed framework(T3V2V) enhance video consistency through self-supervised parameter optimization and domain adaptation.  Authors also provided results which are promising and demonstrates superior spatial-temporal coherence compared to existing models.

## Strengths:

- Relevance: The paper addresses very challenging problem in V2V editing, namely frame inconsistency, which limits the quality and realism of edited videos. The paper is a highly relevant topic to the workshop as well.

- Theoretical Foundation: The paper provides a sound theoretical motivation by analyzing frame inconsistencies through the concept of domain shift, offering a principled approach to the problem.

- Methodological Innovation: The paper introduces a novel TTT method tailored for diffusion-based video generation, including a compute-optimal sampling strategy.

- Promising Results: Experimental results on the DAVIS-EDIT benchmark demonstrate that T3V2V outperforms a state-of-the-art baseline (StableV2V) in subjective and objective evaluations.

- Potential for Generalization: The self-supervised nature of the TTT approach offers the potential for robust generalization to various V2V editing tasks.

## Weaknesses:

- Novelty: While the paper presents a fresh perspective, the individual components are built upon existing techniques, potentially limiting the overall novelty.

- Evaluation Scope: The evaluation is primarily conducted on a single dataset and compared against a single baseline, which could limit the generalizability of the findings.

- Ablation Studies: The paper lacks detailed ablation studies to analyze the contribution of individual components of the proposed framework.

- Clarity: Some aspects of the methodology, particularly the test-time compute-optimal sampling strategy, could benefit from clearer explanations.

- Justification of Choices: The selection of specific pre-trained models and other tools is not sufficiently justified.

- Computational Cost: The paper needs to provide a more detailed analysis of the computational cost associated with the proposed method.

## Areas of Improvement:

- Strengthen Novelty Claims: Clearly articulate the unique contributions and differentiate the work from existing methods.

- Enhance Evaluation: Expand the evaluation to include more diverse datasets and comparisons with a broader range of state-of-the-art models.

- Include Ablation Studies: Conduct thorough ablation studies to analyze the contribution of individual components.

- Provide enough details on disadvantages: It would be helpful understand where the proposed approch might fall behind existing models and this would drive future work.

- Provide Stronger Justification: Offer more robust justifications for design choices, including the selection of pre-trained models.

- Analyze Computational Cost: Provide a detailed analysis of the computational cost and efficiency of the proposed framework.

Overall, the proposed approach by authors addresses complex problem of inconsistencies in V2V editing and proposed approach is demonstrates better results than existing models. Preliminary results gathered using test dataset are encouraging and the subject matter of the paper aligns very well with the workshop's focus. Hence i recommend to accept the paper.

---

### Official Review · Reviewer_FcHv · 2025-03-24
**Review on "T3V2V: Test Time Training for Domain Adaptation in Video-to-Video Editing"**

**Rating:** 2
**Confidence:** 4

**Review:**

Overview:
The paper proposes to use Test-Time Training for Video-to-Video (V2V) Editing. It uses frame-level information to establish an unsupervised TTT learning process and enhances video consistency through effective self-supervised parameter optimization. Experiments on DAVIS-EDIT shows improvement of T3V2V over StableV2V.

Strengths:

1. Good results. The proposed T3V2V shows improvements over StableV2V on all metrics.

Weaknesses:

1. Notations are messy and unclear. To name a few:

(1) The definition of Equations (2) and (3) is unclear. $L(x:\mathbf{I}_t)$ is never explicitly introduced.

(2) $f_{paste}$ in Equations (4) and (8) is unknown.

2. Some claims are tautology. For example, "At the same time, we introduced the concept of spatial understanding to enhance the spatial understanding of the model".

3. Test-time compute-optimal sampling (Sec 3.2 title) is not defined. What does "compute optimal" mean and how is the optimality achieved?

4. Figure 3 needs more explanations in the caption.

---

### Official Review · Reviewer_afaQ · 2025-03-25
**The paper addresses the critical challenge of frame inconsistency (e.g., motion discrepancies, unnatural background changes) in Video-to-Video (V2V) editing models.**

**Rating:** 4
**Confidence:** 3

**Review:**

## Paper Summary
The paper addresses the critical challenge of frame inconsistency (e.g., motion discrepancies, unnatural background changes) in Video-to-Video (V2V) editing. It proposes ​T3V2V, a test-time training (TTT) framework that leverages domain shift analysis and self-supervised learning to enhance temporal coherence. They give a ​domain shift control mechanism to align generated frames with target editing objectives, and a ​test-time compute-optimal sampling strategy for efficient domain adaptation during inference. They integrate ​frame consistency maintenance (via optical flow and object segmentation) and ​spatial understanding (via monocular depth estimation) into the model to guide the Image-to-Video (I2V) generation process. Empirical validation on the DAVIS-EDIT benchmark, demonstrating superior performance over state-of-the-art methods like StableV2V.

​## Strengths
1. ​The integration of TTT with domain adaptation for V2V editing is sound. Existing works focus on fine-tuning or post-processing, while T3V2V dynamically adapts to test-time data. The combination of optical flow-based motion alignment, depth-guided spatial refinement, and gradient-based upsampling is well-motivated. The framework systematically addresses both low-level (pixel consistency) and high-level (scene-object relations) challenges.
2. Quantitative and qualitative comparisons with StableV2V on DAVIS-EDIT are convincing. The visual results (e.g., Figure 4) highlight improvements in detail preservation and temporal smoothness.
3. The proposed compute-optimal sampling strategy balances quality and efficiency, making the method potentially deployable in real-world editing pipelines.

## Weaknesses
1. While the domain shift control is explained, the connection between MSE-based domain shift metrics and the actual video generation dynamics lacks rigorous theoretical justification.
2. The experiments focus on object replacement tasks in DAVIS-EDIT-C. More evaluations across diverse V2V tasks (e.g., style transfer, background modification) and datasets are needed to demonstrate generalizability.
3. The reliance on multiple pre-trained models (SEA-RAFT for optical flow, Florence-2 for frame understanding, Grounding-DINO for segmentation) raises concerns about inference speed and resource requirements. The paper does not report latency metrics or ablation studies on computational costs.

## More Questions
1. How does the proposed MSE-based domain shift metric correlate with perceptual metrics of video consistency?
2.  What is the individual contribution of frame consistency maintenance vs. spatial understanding to the overall performance? Are both modules necessary?
3. Can T3V2V handle long videos (>100 frames)?

---

### Decision · Program_Chairs · 2025-03-25

**Decision:**

Accept

**Comment:**

The paper proposes T3V2V, a test-time training framework addressing frame inconsistency in Video-to-Video editing. Reviewers praised the method's practical relevance, methodological innovation, and strong experimental results. However, they suggested clarifying theoretical explanations, expanding evaluations beyond DAVIS-EDIT, conducting detailed ablation studies, and providing computational cost analysis. The paper is accepted, with authors recommended to address these points in the camera-ready version.